# Living Textures and Mycelium Skin Co-Creation: Designing Colour, Pattern, and Performance for Bio-Aesthetic Expression in Mycelium-Bound Composites

**DOI:** 10.3390/biomimetics10090573

**Published:** 2025-08-29

**Authors:** Anastasia Globa, Eugene Soh, Hortense Le Ferrand

**Affiliations:** 1School of Architecture, Design and Planning, The University of Sydney, 148 City Rd., Darlington, NSW 2008, Australia; 2School of Mechanical and Aerospace Engineering, Nanyang Technological University, 50 Nanyang Avenue, Singapore 639798, Singapore; eugene.soh@ntu.edu.sg; 3School of Materials Science and Engineering, Nanyang Technological University, 50 Nanyang Avenue, Singapore 639798, Singapore

**Keywords:** mycelium, mycelium skin, bio-materials, living materials, sustainable materials, bio aesthetics, MBC

## Abstract

Natural materials present sustainable opportunities in architectural design, but often lack the aesthetic controllability associated with synthetic alternatives. This research explores the bio-aesthetic potential of mycelium-bound composites (MBCs) cultivated from *Ganoderma Steyaertanum* (*Reishi mushroom*), focusing on how external stimuli and surface treatments influence material expression. This investigation was carried out through interdisciplinary collaboration involving design, architecture, and material science. Two post-demolding surface treatment strategies were applied to MBC samples: ‘Delayed Growth‘ and ‘Accelerated Growth‘. These treatments were designed to assess the mycelium’s responsiveness in terms of colour and texture development. A controlled set of samples was analysed using scanning electron microscopy, Fourier-transform infrared spectroscopy, and hydrophobicity testing to evaluate changes in microstructure, chemical composition, and surface properties. The results demonstrate that mycelium exhibits a measurable capacity for aesthetic adaptation, with distinct variations in pigmentation and texture emerging under different treatment conditions. These findings highlight the potential for co-creative design processes with living materials and offer new insights into the integration of biological responsiveness in design practices. The study contributes to the advancement of sustainable material systems and expands the possibilities for bio-design through controlled interaction with bio-materials.

## 1. Introduction

Mycelium-bound composites (MBCs) offer a compelling opportunity to explore how biological systems can inform the development of sustainable, adaptive materials in architecture. This study investigates the capacity of *Ganoderma Steyaertanum* (*Reishi mushroom*) to generate surface-level pigmentation and texture in response to controlled environmental stimuli. By treating mycelium not only as a structural material but also as a living medium capable of aesthetic expression, the research introduces a novel approach to material design that integrates biological responsiveness into the creative process.

The experimental framework was designed to examine how post-demolding surface treatments, specifically delayed and accelerated growth conditions, affect the visual and physical properties of MBCs. These treatments simulate environmental cues that influence fungal behaviour, enabling a form of co-creation between designer and organism. Through microscopy, spectroscopy, and hydrophobicity testing, the study examines how microstructural and chemical changes correlate with visible surface transformations. This research is a pioneering exploration into the controlled aesthetic customisation of mycelium composites, establishing a model for cross-disciplinary collaboration between materials science and design. It contributes to the broader field of biomimetics of materials and built structures by demonstrating how living systems can be harnessed to produce customizable, low-impact materials that respond to their environment. The work advances the field of biofabrication and engineered living materials by proposing novel methodologies for embedding biologically driven adaptive behaviours into architectural systems, enabling dynamic material responses informed by embedded biological adaptability and bio-aesthetics.

This study aims to explore the potential of mycelium, the root structure of fungi, being used as a building material for sustainable architecture, examining mycelium-based material autonomy, aesthetics and performance. Mycelium can grow on bio-waste, is biodegradable, and produces materials with excellent insulating and acoustic properties, along with unique bio-aesthetics [1,2,3]. In the face of an escalating climate crisis and dwindling resources, the need for sustainable materials and the reduction in waste is increasingly becoming a critical issue [4]. Over the past few years, the research focus on sustainable materials has noticeably shifted towards living biomaterials [5,6,7].

Mycelium is an organic material derived from the living body of fungi and produces naturally occurring composites that generate sturdy, lightweight structures that offer unique characteristics for built environment applications [8,9,10,11]. Recent advances in Engineered Living Materials (ELMs) pave the path to novel opportunities for materials science, engineering, and design and have the capacity to surpass the performance and utility of existing smart materials in the near future [12]. Despite ongoing research and multiple start-ups that involve mycelium, the wider adoption of mycelium-bound composites faces significant barriers, with one of the key barriers related to issues with the public response and acceptance [13,14].

This research is a part of a larger research agenda. It builds upon prior investigations into mycelium as a waste-degrading, living-engineered biomaterial, presenting an innovative approach to sustainable material production [15]. The overarching study aims to explore mycelium’s potential as a low-carbon alternative to conventional building materials by leveraging local household waste as a substrate, supporting the circular economy. Several stages of this project were conducted in Sydney, Australia, throughout 2023–2025, with material microscopy testing conducted in Singapore in 2024–2025. The project focused on producing brick-sized mycelium-bound composites using *Ganoderma Steyaertanum* (*Reishi mushroom*) and adopting a low-cost, scalable manufacturing approach. The first research phase examined MBC’s capacity to incorporate materials such as paper, rice, coffee grounds, mandarin peels, and foliage. Each sample underwent growth and inertisation trials to assess mycelium-bound composites’ (MBCs) response to different substrate compositions (Figure 1). The findings of the initial stage underscored the potential and limitations of using local waste-derived biomaterials in architecture, advancing discussions on bio-aesthetic customisation and performance optimisation [15,16]. These findings laid the foundation for follow-up research themes identified as acoustic properties of mycelium composites [17], exploration into programmable material aesthetics and mechanical properties, and investigation of hydrophobic treatments for weather resistance and longevity.

The scope of the research project presented in this paper focuses on controlling and co-creating mycelium-based composite aesthetics, leveraging the Reishi mushroom’s ability to generate pigments and textures on mycelium surfaces under varying conditions. In the initial stage, over 130 test samples were produced, each with its diverse surface characteristics and patterns (Figure 1). In the subsequent research stages detailed in this paper, various systematic surface treatments were applied to a set of 60 additional test samples after demolding and before material inertisation. The results presented in this manuscript clearly illustrate the variety of possible patterning strategies and surface typologies that can be achieved when co-creating with living materials, embracing material autonomy and their reactions to external stimuli and conditions.

The multidisciplinary collaboration between architectural design and material science enabled tests on key properties of the mycelium ‘skin’ using scanning electron microscopy, Fourier-transform infrared spectroscopy, and surface hydrophobicity. This research combines systematic design strategies, material performance evaluation, and mycelium-driven natural biological processes. The findings contribute to the understanding of MBC characteristics, investigating relationships between surface colour, microstructure, composition, and resulting properties. The research demonstrates a scalable and sustainable method for producing mycelium-bound composites with a wide range of natural surface patterns, paving the way for new bio-aesthetics in mycelium-based architectural applications and beyond.

## 2. Background

Visual appearance is the first point of contact between a product and a consumer, user or occupant. The visual appearance often determines a positive response and leads to wider adoption, which many living materials currently lack [18]. The visual appearance of design and architectural products is also key to creating environments that support biophilia and are favourable for the physical and mental well-being of occupants and users. Bioaesthetics can provide a more emotionally pleasing environment by recreating features found in natural environments, such as the use of single or simple colour patterns [19]. In the modern built environment, bioaesthetics can be created by incorporating natural elements, such as plants and wooden panels, or by using paint that mimics natural patterns [20]. However, natural and sustainable building products do not necessarily exhibit the bioaesthetics qualities required for their large-scale consumer adoption and the well-being of users. In particular, mycelium-bound composites grown from fungi onto a lignocellulosic substrate are promising sustainable alternatives for construction material [21], but their public acceptance and commercialisation for these applications are still pending [13,18]. A perception study found that only 56% had heard of mycelium-based composites (MBCs) as decorative materials, and fewer than half were aware of their structural applications. The study suggests that increased familiarity and education could boost professional acceptance of MBCs, especially among architects committed to sustainable design [22]. Aversion to MBCs may also stem from culturally ingrained mycophobia, whereby concerns about fungal growth, mycelium visual appearance and perceived health risks contribute to public hesitancy regarding their use in domestic and consumer applications [18]. There is therefore a need to develop MBCs with a more designable and pleasing visual appearance for their adoption in building designs.

Painting is the first common approach that could be used to alter the visual appearance of MBCs. However, paints are either non-sustainable products containing solvents or are aqueous-based and may lead to humidity absorption by the MBCs. Furthermore, given the visual appearance of panels for the built environments, applying paint is an extra process that comes with an extra cost and negative implications for MBC products’ eventual disposal via natural composting, which is one of the key advantages of unpainted and untreated MBCs. It would be more efficient to find a method where the mycelium itself can change its colour naturally. Indeed, fungal mycelium is a growing living organism that naturally produces pigments, including carotenoids, melanins, polyketides and polyketide-derivatives, and azaphilones [23]. Fungal mycelium of various species has been studied to produce pigments for industries like food, textiles, etc. However, these pigments had to be extracted through processes that break down the fungus. To the best of the authors’ knowledge, there has not yet been any exploration of how to tailor mycelium-bound composite materials’ surface colouration, creating visual patterns for making more appealing and engaging products.

Mycelium hyphae are the vegetative part of fungi that grow underground and are typically white in colour. They are composed of fungal cells that form a 3-dimensional network to connect to the lignocellulosic substrate, which is decayed to provide energy for growth. Despite the white colour of mycelium, some studies have reported alteration of the colour and its browning when exposed to light [24]. The degree of change in colour depends on the exposure type and the type of growth medium. The pigment responsible for the browning is melanin, a large molecule common in fungi that protects it against extreme temperatures, desiccation, UV, etc. [25]. To change the colour of MBCs, they could be exposed to light at various degrees depending on the desired final visual aspect. Changing the light exposure could potentially be a simple, cost-effective and straightforward method to increase the attractiveness of mycelium-based products and increase their public adoption. However, in the initial prototype testing (Figure 1), samples were not exposed to light and were kept in the dark during all growth stages, yet the colour variations were achieved naturally. The only difference that was observed was contact with surfaces and exposure to water/excessive humidity.

In this study, we develop a simple and sustainable process to produce mycelium-bound composites using the fungus species *Ganoderma Steyaertanum* (*Reishi mushroom*) with various natural surface patterns by changing the exposure to different external conditions and non-chemical surface treatments that involve variations in surface contact and hydration levels. This research project investigates how the surface colour is related to changes in mycelium structure, composition and properties. The results of this study demonstrate that the developed approach is scalable for the facile fabrication and design of surface colour MBC bricks or panels for architectural or interior design applications, or any other potential product applications of MBCs.

## 3. Materials and Methods

### 3.1. Aim and Objectives

This research aims to explore the co-creation process between nature and design by investigating the material characteristics of mycelium-bound living materials using *Ganoderma Steyaertanum*. It focuses on developing scalable and sustainable methods for designing surface colours and patterns to enhance bio-aesthetic appeal and controlled performance in architectural applications of mycelium-bound composites.

Objective 1: Controlled Surface Design. Investigate how different surface contact treatments and external applications influence the MBC skin colour, patterns, and texture of mycelium-bound living materials to achieve desired bio-aesthetic qualities.

Objective 2: Material Autonomy and Growth Response. Examine the extent of mycelium’s autonomous behaviour under varied conditions, assessing variations in its growth colours, patterns and responses to applied treatments.

Objective 3: Structural and Mechanical Properties. Analyse the mechanical and hydrophobic/hydrophilic differences between varying skin colours and textures of mycelium skin, identifying how surface characteristics affect performance and potential applications.

Main Research Question: To what extent and how can the skin/surface colour, patterns and texture of the mycelium-bound living materials be controlled or designed using different mechanical (non-chemical) skin surface treatments and applications to achieve desired bio aesthetics and controlled performance?

Sub-Questions: How does mycelium, as a living material, exhibit self-directed growth and autonomy when subjected to various conditions and treatments across different samples? What are the mechanical and structural differences between different skin colours and the texture of the mycelium skin?

**Hypothesis:** *Mycelium-based living materials derived from Ganoderma Steyaertanum mushroom display distinct colour and texture changes depending on surface treatments and environmental conditions during growth. These factors influence the development of the mycelial skin, potentially causing yellowing, browning, or the formation of fruiting bodies (visible mushrooms), which marks a shift from vegetative to reproductive growth. We hypothesise that applying mechanical surface treatments can help control these outcomes to achieve the desired material properties*.

### 3.2. Methods

This research project progressed as an iterative experimental study. Three rounds of testing were performed to explore and refine research hypotheses. Initially, it was observed that the MBC samples’ skin tended to grow darker at the locations where the plastic wrapping was touching it after the demoulding stage (Figure 1). Over 130 samples were produced during this stage, and this response appeared to be a consistent trend. Water condensation collected on the surface of the plastic wrap that was introduced to control humidity was naturally transferred to the locations where the plastic wrap was touching the MBCs samples’ skin surface. Therefore, our initial hypothesis was that excessive condensation or contact with water accelerated the growth of hardened mycelium skin and led to the formation of yellow discolouration on its surface.

The selection of *Ganoderma Steyaertanum*, commonly known as the *Reishi mushroom*, was informed by a series of controlled experiments conducted by our research team between 2019 and 2025, aimed at evaluating various mycelium species for architectural applications. Among the commercially available strains in Australia, *Ganoderma Steyaertanum* demonstrated superior resilience to mould and contamination, and consistently formed a protective skin over the substrate, which is an advantageous trait for material stability for the built environment applications. This performance was significantly better than other species tested, including Grey Oyster, Winter Chocolate Oyster, White Oyster, Blue Oyster, and Lion’s Mane mushrooms. The duration and environmental conditions of the growth and treatment cycle were similarly based on empirical data from these prior studies [9,16], which established optimal parameters for consistent living material growth and surface characteristics.

### 3.3. Experimental Set-Up

#### Materials and Sample Preparation

For this exploration the constant conditions were set as follows: substrate type and composition—woodchips inoculated with *Ganoderma Steyaertanum*—commercially available at Aussie Mushroom Supplies [26], room/growth temperature, dark space—no access to light, volume/weight/shape of the samples (750 mL plastic containers), moisture levels, length of growing cycles (Figure 2).

The changing variables included the shape, coverage, and detailing of the skin treatments, as well as the type of treatment material (such as plastic, wood, sponge, clay, metal, etc.). Three rounds of tests that iteratively informed each other. The growth and treatment cycles followed the 16–17-day cycle: 14 days of growth inside the mould, constant for all samples and 2–5 days of growth after demoulding, allowing the skin growth of MBC samples. During this stage, various treatments and applications were combined. Samples were split into different test groups to explore various types of treatments. For each iteration stage, twenty 750 mL plastic containers were prepared, each fitted with two 3 cm ventilation holes on opposite sides of the container and sealed with breathable tape to ensure airflow and access to oxygen but minimise the loss of moisture. Alternatively, this could also be achieved by using perforated plastic wrap. Mycelium needs oxygen to grow, so providing consistent access to air and ventilation while in the mould was an important consideration. Adequate levels of moisture are also critical for mycelial growth and nutrient uptake, as insufficient hydration would impair substrate colonisation. The use of breathable micropore tape and perforation allowed both access to oxygen and prevented rapid water evaporation from the substrate. Additional misting with water was applied every 3rd to 4th day to keep the moisture levels consistent.

During the mould-filling process, all equipment and surfaces were sanitised with antibacterial spray, and gloves were worn throughout to maintain sterility and prevent potential contamination. Each container was filled with 350 g of substrate sourced from commercially available Reishi mushroom grow kits [26]. The samples were then placed in a dark environment with daily room temperatures ranging from +15 to +23 °C and humidity of 45% to 85% to support consistent conditions (Figure 2).

### 3.4. Iterations 1 and 2

Twenty samples were produced for the first and second iterations. For these two initial iterations, the approach was to apply laser-cut plastic (Polypropylene 1 mm sheets) cut-outs after the demoulding stage onto the top surface of the mycelium samples to simulate conditions that occurred naturally with plastic wrap touching the surface and triggering different skin colour responses (Figure 1). The aim was to do so in a more controlled and rigorous way (Figure 3). To investigate this systematically, the cut-outs were designed following a simple-to-complex geometry logic. The samples were split into 5 test groups with 4 samples in each group. Group samples are illustrated in Figure 3: Group A ‘ON/OFF’, Group B ‘Gradient’, Group C ‘Patterns’, Group D ‘Shapes’, Group E ‘Detailing’.

Following the growth phase (14 days), samples were carefully removed from their containers and placed into custom digital skin test cutouts to ensure full contact between the fungal material and the test surface, to observe any variation in skin growth colours, where the plastic was touching the mycelium skin surfaces vs. exposed surfaces. Samples were left for another 3–5 days to allow the formation of a fungal skin layer. Visual changes in colour and surface texture were documented throughout this period. Subsequently, samples underwent a controlled drying process in a fan-assisted oven at 50 °C for six hours to halt mycelial growth and follow-up air drying for another 7 days, to ensure complete intertisation of the MBC samples. In these two initial test iterations, the application did not provide consistent results, contrary to the expected outcomes. Thus, failing or partially failing to prove the hypothesis (Figure 4 and Figure 5). Several key findings and observations were made that could have prevented the desired results. These were recorded to inform the design and set-up of the follow-up iterations.

### 3.5. Iteration 1: Lessons Learned

The experimental process highlighted several challenges related to substrate uniformity, moisture control, and cutout design during mycelium skin growth. These included irregular surfaces caused by coarse, large-size (over 1 mm) substrate particles (e.g., sawdust and wood chips) impaired contact between the fungal material and test cutouts. To address this, in the follow-up iterations, a more refined substrate with large particles removed or blended for greater homogeneity was used. Additionally, early discolouration of the mycelium from white to yellow was observed, likely due to over-misting and substrate water saturation. Improved moisture management, including reduced misting and more even distribution during initial growth stages, was adopted to maintain healthy mycelial growth. Lastly, a longer growth period after demoulding was identified as another area of improvement, shifting it from three to five days in the follow-up iterations.

### 3.6. Iteration 2: Lessons Learned

This second iteration of sample testing and the experimental setup addressed the identified issues, such as surface unevenness and excessive condensation during mycelium skin growth. Key improvements included substrate refinement and enhanced mould design to promote better adhesion between the mycelium and treatment materials, particularly plastic cutouts. In addition to the twenty samples, the study also explored alternative treatment materials beyond polypropylene, including more porous/water-absorbent materials such as sponge and clay, resulting in four additional samples.

Twenty-four samples were produced for this test. In this iteration, the substrate was blended to a fine consistency to eliminate large particles and ensure a smooth surface top. Containers were kept in a dark space for two weeks, with daily monitoring of temperature and humidity. After that, all samples were demoulded and treated with the designated cut-outs and additional test materials. They were then left to grow the skin layer after demoulding for five days. Visual changes in colour and texture were recorded throughout. Finally, samples were dried to stop growth and stabilise them for further analysis.

The second iteration of the study showed more consistent results (Figure 5), and the findings from this test stage informed the final iterations of this investigation, helping to articulate and develop two alternative approaches to co-creating colours and patterns on the surface of MBCs. As illustrated in Figure 5, Groups A and B produced the most notable results. Group A confirmed that increased surface contact (0–100%) correlates with more pronounced skin development, particularly dense brown skin in high-contact areas. Group B showed that gradient patterns led to localised skin darkening where contact was greatest. In contrast, Groups C and F showed no significant results, suggesting insufficient contact or incompatible materials. Group D revealed that denser, smaller contact points promoted more dramatic colour change. Groups that explored porous and dense materials, respectively, showed mixed outcomes. However, it was observed that for the samples where weighted treatments were applied, the changes in skin colour were especially evident.

These findings informed the design of follow-up tests in several key ways. First, the results confirmed that consistent and firm surface contact is critical for inducing mycelium skin colour change, particularly the denser, brown variant. This insight led to the decision to replace flexible plastic treatments with more rigid and heavier materials in subsequent experiments to ensure uniform pressure distribution. Second, the limited response to tested porous materials emphasised the importance of using more rigid contact surfaces to achieve the desired response. The follow-up tests focused on using more solid materials such as plastic and metal.

It was observed that the access to oxygen and the amount of moisture/water on the skin surface of mycelium (and subsequent treatment capacity to delay or accelerate the rate of water evaporation) had a pivotal effect on the resulting colour of the skin. Lastly, it was observed that initially, the mycelium skin grows as white colour, as time progressed and depending on the amount of moisture/humidity 1–3 days later, the white colour is likely to shift to yellow or yellow/orange hue, after day 4–5 or when excessive water is present on the mycelium skin surface the colour is likely to darken further, shifting to light or dark brown. As a result of these preliminary test iterations, two alternative approaches to surface treatment were developed: Delayed Growth and Accelerated Growth. These were explored in the final, third iteration described below.

## 4. Implemented Approaches and Evaluation

### 4.1. Approach #1 Delayed Growth/Maturing

During the first iterations, it was observed that continuous and uniform contact between the mycelium substrate and mould surfaces consistently prevented the formation of the mycelial skin layer. This happens most likely due to either the lack of access to oxygen or the lack of access to moisture. It appeared to delay the mycelium’s natural surface development, particularly in areas where mould contact was constant. More growth and skin colour change were observed at the locations where ‘breathing’ openings were introduced into the moulds. These findings suggested that the presence or absence of surface contact plays a critical role in the morphological development of mycelial skin.

We hypothesised that by applying targeted and patterned surface contact, rather than full or continuous coverage, we can intentionally delay the onset of skin formation in specific regions. This controlled delay may influence not only the timing of skin emergence but also the colouration and texture of the resulting fungal layer. In particular, we anticipated that areas under delayed exposure to air and moisture would exhibit distinct visual and structural characteristics (be whiter) compared to earlier exposed zones (be more yellow/brown). By manipulating the timing and intensity of contact, we aim to observe how delayed interaction affects skin development. Such outcomes could provide valuable insights into the skin growth dynamics of mycelia and open new possibilities for designing living materials with co-created controlled properties.

To test this ‘Delayed Growth’ approach, twenty samples were produced. Laser-cut weighted plastic cutouts were applied following five established test group treatments (Figure 3). As with previous iterations, the ‘in-mould’ growth was set for 14 days. These cutouts varied in shape, contact area, and size, and were applied immediately after demolding. For three days, they were held in place. After day three, the cut-outs were moved and applied again (secondary delay) at a different angle and left for another 2 days. Key variables to be controlled include contact timing (3 days vs. 5 days), contact duration (short-term vs. sustained), pattern geometry and density (5 test groups, simple/sparse vs. detailed/dense).

We expect that delayed access to oxygen and moisture facilitated by the presence of the cut-outs will result in a delayed cycle of skin darkening, with treated areas showing postponed or altered skin maturity (Figure 6). These regions would exhibit colour differences (e.g., delayed yellowing/browning) and texture differences (e.g., softer or thinner layers). The samples from iteration 3 were used to test both approaches #1 and #2, where approach #1 treatments were applied to the opposite side of the samples.

### 4.2. Approach #2: Accelerated Growth

In earlier phases of the study, it was observed that direct contact with water or persistently moist surfaces could accelerate mycelial growth and lead to the formation of darker, denser skin patterns. This effect was particularly noticeable in areas where evaporation was restricted, suggesting a relationship between sustained hydration, oxygen deprivation, and altered fungal behaviour. This is likely due to the mycelium’s skin defensive mechanisms, rapidly increasing the thickness of the skin (from soft white to firm brown) to protect its main body from increased hydration, which may shift the metabolic activity of the mycelium and influence pigmentation and texture. It was also noticed that in some cases, the excessive amount of water could potentially lead to either fruiting (formation of the mushroom reproductive organ) or possible failure and mould growth.

We hypothesised that applying heavy moisture-retaining/trapping materials to the surface of already-formed white mycelium skin can induce localised darkening and densification of the mycelium skin. This hypothesis formed the core of the second approach—‘Accelerated Growth‘ (Figure 7). Unlike approach #1, which systematically closed the exploration loop for iterations 1 and 2, approach #2 was intentionally exploratory, aimed at testing a wide range of water-retaining shapes, contact surfaces and material interactions on already fully formed mycelium skin layer, with white mycelium skin fully formed. Due to the size limitations of the sample moulds, it was not possible to apply the same plastic cut-outs from iterations 1 and 2 (Figure 3) to test approach #2. Various shapes and materials were applied to the surface of demoulded mycelium samples, including metal, metal wire, cork, sponge, and wood. These materials were either pre-moistened or used in combination with water to create localised high-humidity zones. For this approach, the top surface of the samples was left exposed, not touching the mould directly, to allow gradual growth of the skin layer (11 days).

Initially, the top surface did not form the skin layer actively as the sample substrate was not fully colonised yet. Mycelium skin growth on the exposed surface was relatively slow in the first 7–10 days. After 11 days, the top layer of the samples became white, forming the initial skin layer. Treatments were applied directly to the top fungal surface and left undisturbed for three days to observe changes in colour and texture. After 3 days with approach #2 used, the surface treatments were removed, and the samples were flipped over to ensure sustained contact with the mould surface, aiming to prevent or slow down further growth (limiting exposure to oxygen and water). The inertisation occurred on day 20; therefore, the samples had a total of 8 days post-approach #2 application and before the growth was halted completely (Figure 7).

### 4.3. Evaluation of Results: Characterisation and Testing

The macro photography for this study was conducted using a studio light box (505 × 485 × 495 mm) equipped with adjustable colour temperature lighting (3000–5600 K, set to 5000 K) and a luminous flux of 3500 lumens. Photography was performed using a multi-camera system comprising three integrated lenses (50 MP, 10 MP, and 12 MP) with apertures of F1.8, F2.4, and F2.2, respectively. The system employed autofocus and HDR capabilities, with image processing algorithms optimising focus, depth and low-light performance. Figure 8 illustrates a representative sample produced during the initial testing phase, showcasing the full spectrum of mycelium skin colour variation achievable using the *Reishi mushroom* (*Ganoderma Steyaertanum*) substrate used in this study. Based on observations from a total of 190 samples, the colour gradient spans from nearly white (category (1), through yellow (2), orange (3), and brown (4), to, in rare instances, almost black (5). Each of these five distinct skin colour categories was individually analysed to gain deeper insight into the unique properties and characteristics associated with each variation. The testing procedures conducted in this study included electron microscopy, compositional analysis, and surface hydrophobicity measurements. However, due to limitations in sample size, flatness and the specifications of the available equipment, hardness testing, specifically Vickers hardness and indentation fracture toughness (KIC), could not be performed as initially planned. This constraint highlights an area for future investigation, which could be addressed in follow-up studies to provide a more comprehensive understanding of the material’s mechanical properties.

#### 4.3.1. Electron Microscopy

Electron micrographs were obtained using a scanning electron microscope (SEM, JSM-5510LV, JEOL, Akishima, Japan) on samples deposited onto carbon tape and coated with gold for 45 s (Cressington 108 Gold Sputter coater, Watford, UK). The gold coating was necessary to make the samples electrically conductive for their observation in the SEM. Small pieces of the different regions of the mycelium skin were cut out using a scalpel and subsequently dried in an oven (IKA-48, IKA Works (Asia) Sdn Bhd, Selangor, Malaysia) at 48 °C overnight prior to observation under the SEM.

#### 4.3.2. Compositional Analysis

Fourier-transform infrared spectroscopy (FTIR) was conducted with an FTIR Spectrometer (PerkinElmer Frontier FT-IR Spectrometer, Waltham, MA, USA) using the Attenuated Total Reflectance method. Small samples of the different regions were cut out using a scalpel. Each sample was measured thrice at different areas of the samples in the 4000 to 800 cm^−1^ range with 4 cm^−1^, accumulating 32 scans. Transmittance values were recorded in triplicate and then averaged and are reported in the form of mean ± standard deviation.

#### 4.3.3. Surface Hydrophobicity

The hydrophobicity of the samples was measured by dropping 10 μL of deionised water on selected smooth surfaces of the different regions of the sample. An optical camera was set up to capture the images of the water droplet over a 5-min period at every 1-min interval. ImageJ Version 1.54p was used to measure the angle of the water droplet and the fungal skin. This process was repeated three times to obtain an average, and the results are reported in the form of mean ± standard deviation.

## 5. Results

### 5.1. Approach #1: Delayed Growth/Maturing

Approach 1 tested a hypothesis that targeted contact between pattern cutouts or surfaces and the outer layer of the mycelium-bound composite (MBC) could delay the growth and pigmentation of the mycelium skin by restricting access to oxygen and moisture. Figure 9 presents the results from the third and final iteration of testing using this ‘Delayed Growth‘ method. Drawing on insights from the first two iterations, the experimental setup was refined to ensure a flat and even top surface of the MBC sample. To enhance adhesion and maintain consistent contact between the MBC surface and the applied cutouts, weights were used to secure the cutouts in place. This adjustment improved the reliability and consistency of the delayed growth effect observed in the treated areas.

The upper part of Figure 9 shows the day-by-day progression of skin growth and colour change in samples C2 and C4, from day 1 to day 5. The colour transformation throughout these 4 days ranges from white to dark brown. A clear difference can be observed between the areas covered by the transparent plastic cutouts and those left exposed. On day 1, both samples showed minimal skin growth. By day 2, extensive white-coloured skin growth appeared in the exposed areas, while growth under the covered regions remained limited, indicating an expected skin growth delay. On day 3, the exposed areas began to shift in colour from white to yellow/orange, while the covered areas continued to show slow, partial growth with a white appearance. At the end of day 3, the cutouts were slightly repositioned to examine the effects of partial and prolonged exposure. On day 4, the previously exposed areas darkened further, turning light brown, while the newly exposed regions remained predominantly white, with some areas beginning to show yellow tones. By day 5, the exposed surfaces had darkened to deep brown, while the covered areas retained white, yellow or orange colouration.

The bottom section of Figure 9 presents all samples tested under Approach 1. Across the full set, results were consistent, tangibly proving the effectiveness of the proposed method. In some cases, particularly where the surface of the MBC samples exhibited slight dips or unevenness, the intended response of the applied patterns was not fully achieved. This limitation is evident in samples B1, B2, D1, E1, and E2. Smaller (<1 mm) design elements were especially susceptible to surface irregularities, as seen most clearly in samples B2 and E2. Samples A1 and A3 demonstrate that the desired colour contrast effect can be successfully achieved on larger, more uniform surfaces, as part of the On/Off test group. Samples B3 and B5 further illustrate the potential of this method to generate secondary visible patterns through the repositioning of the cutouts (Figure 9). However, in most cases, the areas that were originally exposed continued to darken even after being covered on day 3, suggesting that initial exposure had a lasting impact on pigmentation.

Overall, the results from this experimental stage confirm that the “Delayed Growth” approach yields consistent and reproducible outcomes, thereby validating research Hypothesis 1.

### 5.2. Approach #2: Accelerated Growth

Approach 2, ‘Accelerated Growth‘, explored the hypothesis that darker pigmentation in mycelium skin can be achieved through targeted surface contact combined with increased local skin hydration. Specifically, it was proposed that when white mycelium skin has already formed, maintaining high levels of moisture, either by preventing evaporation or by applying water-saturated cutouts or surfaces, would result in darker colouration in the treated areas. The main differences from the first approach were that the contact surfaces were pre-wetted before applying the contact shapes, and the treatments were applied to mycelium skin that had already fully formed and turned white.

As shown in Figure 10, the hypothesis behind Approach 2 was successfully validated, although the results were less consistent and uniform compared to those observed in Approach 1. One of the limitations of this study is that a direct comparison between these two approaches cannot be reliably made, as they utilised different surface treatments and materials. However, as proof of concept, both delayed and accelerated growth approaches were successfully tested and validated.

The lower portion of Figure 10 demonstrates that mycelium skin generally exhibited colour changes when exposed to wet surfaces. This effect was most pronounced when heavier, non-porous objects were applied, as seen in samples 2, 8, 9, 10, 11, 12, 14 and 15. In contrast, materials such as cork and sponge produced less noticeable colour changes but were significantly more difficult to remove due to the mycelium bonding with and growing into these porous materials (samples 5, 13, 16 and 19). Natural materials such as timber (1), cork (13) and foliage (20) were partially colonised by the mycelium over the three-day application period, making removal challenging as well. Among these, wood treatment led to a distinct colour change, while cork produced a more subtle contrast. Notably, samples 7 and 20 displayed deeper red and dark brown tones on the exposed skin, whereas samples 4, 8, 14 and 15 showed lighter browning in the treated areas.

Lighter or less dense objects (Table 1), as well as those with uneven surfaces, tended to produce less consistent pigmentation effects, as observed in samples 3, 4, 5, 7, 13, 16, 17 and 19. This inconsistency is likely due to poor surface contact, which allowed significant moisture to evaporate from the treatment areas, reducing the effectiveness of the hydration-based approach. In contrast, sample 18, which tested the direct application of water droplets to the mycelium surface, showed promising results. The skin darkened quickly and uniformly under this treatment, indicating strong potential for water-only applications in controlled mycelium skin pigmentation.

The upper section of Figure 10 illustrates that the ‘Accelerated Growth‘ approach yields less consistent results in producing darker pigmentation beneath the treated areas, as seen in samples 2, 9 and 10. In some cases, the skin darkened to brown within three days; however, there was also a significant likelihood of these areas remaining white at the same stage (Figure 10 and Figure 11). Over extended growth periods, such as by day 8, the treated regions tended to darken further. A notable limitation of this approach is that the exposed mycelium skin also undergoes natural darkening over time, which complicates efforts to achieve precise and reliable colour differentiation in the treated zones.

### 5.3. Findings and Observations

The experimental results from both Approach 1 (‘Delayed Growth‘) and Approach 2 (‘Accelerated Growth‘) demonstrate that mycelium skin pigmentation and surface characteristics can be effectively manipulated through targeted environmental interventions. Each approach yielded distinct outcomes, with varying degrees of consistency and control (Figure 9, Figure 10 and Figure 11). Approach 1 successfully delayed mycelium skin growth and pigmentation by limiting oxygen and moisture exposure through the application of surface cutouts. This method produced highly consistent results across multiple samples, with clear contrasts between exposed and covered areas. The pigmentation followed a predictable gradient from white to yellow, orange, and brown, depending on the duration of exposure. This approach allowed for precise patterning and spatial control of colour development.

Approach 2, which involved increasing local surface hydration on already-formed white mycelium skin, also proved effective in inducing darker pigmentation. However, the results were less uniform compared to Approach 1. Darker tones, including deep brown and even black, were more likely to occur under this method, particularly when heavier, non-porous materials were used to maintain moisture. Interestingly, a recurring phenomenon was observed in which a white outline often formed around darker regions, suggesting a form of material autonomy or self-regulating growth behaviour in response to localised external changes (Figure 11, sample 15). Another notable observation across both approaches was the white-to-dark gradient that frequently appeared near areas of high pigmentation. This gradient effect was especially prominent in Approach 2, where the transition from hydrated to non-hydrated zones created a natural diffusion of colour intensity. The occurrence of black pigmentation, rare in Approach 1, was more commonly observed in Approach 2, indicating that prolonged hydration may trigger deeper biochemical changes in the mycelium skin.

These findings suggest that not only can surface treatments influence aesthetic outcomes, but they may also reflect underlying biological responses and autonomy of the mycelium-bonded material. The emergence of gradients, outlines, and autonomous patterning points to the potential for self-organising behaviours in fungal composites, which could be further explored in future studies.

### 5.4. Relationships Between Surface Colour, Microstructure, Composition and Properties

The change in colour of the mycelium skin could be the result of changes in the composition and microstructure of the mycelium layer underneath. These chemical and structural changes would likely result in changes in properties. To investigate these aspects, mycelium skins of different final colours were evaluated.

Electron microscopy images show the morphology of the surfaces of the different regions at magnifications of 500× and 2500× in the middle and right columns, respectively (Figure 12). At the 500× magnification, all images show that the mycelium network has formed a cohesive network that forms the fungal skin. Pores are more easily identifiable in the yellow and red/orange region, as the surface morphology seems to be rougher and less uniform. At a closer inspection, the shape and structure of mycelium hyphae growth differ with respect to their different regions. In the white region, where areas are left untouched, the mycelium hyphae form coral-like structures, with short hollow tubes growing in all directions. While the pores were not observable at the 500× magnification, minute pores between the hyphae structure can be observed here. In the yellow region, thread-like mycelium hyphae are observed, similar to observations in the literature [27].

Pore sizes are observed to be much bigger between the gaps of the mycelium network. These thread-like mycelium hyphae seem to become swollen in the red region, where ball-like structures are also observed. An increase in width in the mycelium hyphae is generally observed in the red region. This could be the result of some contact on the mycelium skin by the mould, where the additional pressure prevents the mycelium network from growing in all directions and instead causes it to swell. In the brown and black regions, the mycelium hyphae network has fused, forming a coating that has little to no pores. The increase in weighted contact causes the mycelium hyphae to not only swell but to start fusing together, forming a skin-like coating.

The darker regions tend to have flat and compacted hyphae networks, where it is unable to grow upwards due to contact with the mould or weight pressing down on it. These observations indicate that the darker regions are smoother, while the lighter regions tend to be rougher on the surface, which could impact the hydrophobicity of the surface of these different regions. The different growth structures could also indicate the autonomy of the growth behaviour of the mycelium hyphae network. FTIR was then used as a characterisation technique to identify any underlying biological responses that lead to the fungal skin being able to develop different hues.

ATR-FTIR spectroscopy was used to characterise the chemical nature of the different regions of the fungal mycelium skin (Figure 13 and Figure 14). The infrared absorption spectra of the mycelium hyphae are associated with the biomolecules that compose them, where the main absorption of lipids (asymmetric and symmetric CH2 stretching modes at 2935), proteins (amide I and II at ∼1645 and ∼1545 cm^−1^, respectively), and chitin (CH bending mode at ∼1375 cm^−1^), polysaccharides (CC stretching mode at ∼1020 cm^−1^), are identified [28]. A lower transmission value indicates that less light is transmitted through the sample and thus possesses a larger amount of the particular functional group. Generally, a split between the lighter and darker hues can be seen where they are similar in trends. For the lighter regions, a higher peak intensity is observed generally throughout the spectra, especially at the 4 major wavelengths, as compared to the darker regions, indicating an increased presence of these 4 functional groups. Between the 1000–1700 cm^−1^ range, the white region is observed to contain an increased number of proteins, lipids and chitin compared to the other regions. However, at the 1020 cm^−1^ mark, there is a significant spike in polysaccharide content for the yellow region, which suggests a larger concentration of cell structures or mycelium hyphae as compared to the other regions.

Based on the spectra, three infrared ratios, namely lipid/polysaccharides, protein/polysaccharides, and chitin/polysaccharides, were calculated and characterised to quantify the presence of functional groups. Due to the lack of difference in structures, the ratios of the functional groups followed similar trends among the different regions, suggesting that while the surface morphology has changed, the presence of functional groups has not been greatly affected. Different characterisation techniques could be explored, such as chromatography, which could help provide a better analysis of the proportion of functional groups present. The different regions, as identified by their hues, could also be characterised by the presence of melanin, which could be the cause of the different tones present.

The results provide an insight into the chemical composition of the functional groups present at the surface of these different regions and can serve as an understanding of the properties of the various regions. To further characterise the functional properties, the hydrophobicity of these regions was tested.

Optical images of water droplets deposited onto the surface of the different regions were taken over a 5-min period at minute intervals (Figure 15). As per literature, the contact angle found in mycelium composites is typically reported at 120–130° and also possesses the ability to maintain hydrophobicity over extended periods [29]. White and yellow regions demonstrate initial hydrophobicity with contact angle values of 116° and 113°, respectively. The contact angle in the yellow region was sustained throughout the 5-min period. A slight decrease was observed in the white region, which could be caused by the bumpy nature of the surface where the water droplet slips, causing the contact angle to decrease and eventually stabilise. When the contact angle falls below 90°, the surface is known to be hydrophilic, where the droplet spreads onto the surface and becomes wetted. This is observed with the darker orange and brown regions, where the initial contact angle is measured at 82° and 65°, respectively.

In addition, the contact angle continues to decrease over time, indicating that the surface of these regions is not hydrophobic and is wetted over time. There are no results for the black region due to the bumpy surface, where no flat areas could be used for contact angle measurement. The surface morphology as seen through the electron microscopy images could present an explanation for the difference between the hydrophobic nature of the lighter and darker regions, where the increase in surface roughness in the morphology increases the surface energy and allows the water to remain in its droplet form. In comparison, the smooth surface does not provide sufficient surface energy and thus allows the water to slip and eventually causes the surface to be wetted.

These results demonstrate that the change in colour is associated with changes in surface morphology, composition and properties. The mechanical properties could not be tested and may also depend on the mycelium skin thickness. The main result obtained is that the mycelium surface loses its porosity and becomes less hydrophobic.

## 6. Discussion

Visual appearance plays a critical role in the acceptance of materials within architecture and design. Mycelium-bound composites (MBCs), despite their sustainability, often lack the bioaesthetic qualities needed for broader adoption [18,19,20,21,22]. Conventional methods to enhance appearance, such as painting, introduce sustainability and cost concerns, particularly for compostable materials [20]. Our study offers a feasible and sustainable alternative by demonstrating that natural colour variation in MBCs can be achieved through controlled environmental conditions, specifically surface contact and hydration, without any need for chemical additives.

The two approaches explored in this study, ‘Delayed Growth‘ and ‘Accelerated Growth‘, demonstrate viable methods for controlling and co-creating the surface characteristics of mycelium skin, enabling the desired development of textures and pigmentation. Gathering more comprehensive data on the materials used to create the surface colours, such as their weight, contact pressure, porosity, temperature, contact area, and dwell time, could be further implemented to correlate those properties to the achieved colours. These findings open new possibilities for design-driven applications of fungal materials, particularly in fields such as sustainable architecture, product design, and bio-fabrication. While the ‘Accelerated Growth‘ approach produced promising visual outcomes, we acknowledge variability in colour consistency across samples due to several factors such as moisture application technique and the physical properties of contact materials (e.g., weight, density, and flatness). These variables influenced mycelium skin colour development and surface uniformity. To improve reproducibility, future experiments will incorporate contact materials with controlled weight, contact area and flatness, and apply water using calibrated methods to regulate volume and exposure time.

Future research could aim to refine these techniques by systematically controlling key variables such as moisture levels, contact duration, and exposure to oxygen and humidity. Additionally, biochemical and structural analyses of light versus dark skin regions will be essential to uncover the underlying mechanisms driving these visual and material transformations. Methods such as high-performance liquid chromatography (HPLC), Raman or UV/Vis spectrophotometry, or targeted melanin assay could be used to quantify the exact content of melanin, carotenoids, or other molecules and understand the mechanisms for the formation of colour on the mycelium. While this study focuses on colour control and transformation, longer-term growth, skin darkening/maturing effects remain an open area for investigation. Preliminary observations suggest that environmental factors such as humidity and UV exposure may also influence colour formation during the growth stage. Future work will explore methods to control or stabilise these changes, potentially through surface treatments or environmental conditioning. The use of adhesive surface applications can also be explored to maximise the application effect/accuracy.

Beyond the current scope, several promising directions warrant further exploration. One direction involves the use of natural coatings (during post-intertisation stage), such as plant-based oils or waxes, which may influence pigmentation in a manner similar to wood oiling, making colours more vibrant and contrasting. These treatments could offer both aesthetic and functional benefits, including enhanced durability or water resistance, while still allowing the resulting MBCs to be fully compostable and sustainable. Expanding the range of material treatments and cut-out materials also presents opportunities for more nuanced control over skin development. Investigating how different contact materials, porous, non-porous, organic, or synthetic, interact with mycelium could lead to more predictable and customizable outcomes. The size, types and composition of mycelium substrates could also affect the appearance and colours, as it was observed in previous research that substrates containing coffee grounds tended to produce darker samples, for example [15]. Future research might aim to explore whether, and to what extent, the type and amount of biomass used to grow the mycelium contribute to the colour change process from white to brown.

Another area of interest that was beyond the scope of this research project is the reaction of mycelium skin to light and sunlight, particularly in terms of pigmentation changes, UV sensitivity, and potential photoreactive properties. Understanding how fungal materials respond to light exposure could inform future applications in outdoor environments or dynamic surface treatments.

Further investigation is needed to assess the structural integrity and weather resistance of mycelium skin in relation to its surface colour and texture. Preliminary observations suggest that pigmentation and surface characteristics correlate with differences in density, moisture retention, and surface textures, which could influence the material’s performance under environmental stress. More extensive testing, such as tensile strength, abrasion resistance, water permeability, and UV exposure, could help determine whether darker or more textured regions of mycelium skin offer enhanced durability or resilience. Understanding these relationships and the possible correlations between colour–structure–performance would be particularly valuable for applications in outdoor environments or products requiring long-term stability. Such insights could also inform the development of functional grading strategies, where surface modulation is used not only for aesthetic purposes but also to optimise material performance across different zones of a single composite.

Finally, testing these approaches across different mushroom species and substrate compositions will be critical for generalising the findings and improving the adoption of proposed colour control methods. Variations in fungal biology and growth behaviour may yield diverse aesthetic and structural results, broadening the applicability of these techniques across a wider spectrum of biofabricated materials. Collectively, these directions point toward the development of responsive or programmable fungal materials, where environmental inputs, such as moisture, air, light, and surface contact, actively guide the aesthetic and functional properties of the material. This line of inquiry holds significant potential for advancing sustainable material innovation and expanding the creative possibilities of mycelium-based design.

## 7. Conclusions

This study demonstrates that the visual characteristics of mycelium-based composite (MBC) skin appearance and colour can be effectively controlled through two distinct strategies, delayed and accelerated growth, each yielding unique and reproducible pigmentation outcomes. The ‘Delayed Growth‘ approach, which restricts oxygen and moisture through surface contact, consistently produced high-contrast patterns and controlled colour transitions from white to deep brown over five days, validating the hypothesis with strong visual differentiation. In contrast, the ‘Accelerated Growth‘ method, involving hydration of pre-formed white mycelium skin, showed promising but less consistent results, with darker pigmentation emerging under heavier, non-porous materials and direct water application. Limitations such as surface irregularities and material porosity affected the uniformity of outcomes in both approaches. Nonetheless, the findings confirm that MBC skin pigmentation can be directed with precision, offering architects and designers a novel method for aesthetic customisation using sustainable biomaterials.

Both approaches revealed autonomous patterning behaviours, such as white outlines and gradient transitions, suggesting self-regulating responses within the fungal network. Microscopic and spectroscopic analyses further demonstrated that pigmentation changes correspond to distinct structural and chemical transformations in the mycelium hyphae. On the microscopic level, lighter regions exhibited coral-like, porous structures with higher concentrations of proteins, lipids, and chitin, while darker regions showed compacted, fused hyphae with reduced porosity and increased surface smoothness. Hydrophobicity tests revealed that lighter regions retained water-repellent properties, with contact angles of 116° (white) and 113° (yellow), while darker regions became hydrophilic over time, with angles dropping below 90°, indicating wettability. These changes in surface behaviour are linked to differences in surface roughness, suggesting that pigmentation is not merely aesthetic but indicative of deeper material transformations.

Future research should explore how these changes affect mechanical performance, durability, and environmental resilience, particularly in architectural applications. Additionally, expanding the study to include different fungal species, substrates, and post-treatment methods, such as natural coatings or light exposure, could unlock new possibilities for programmable, sustainable design materials.

These results contribute to the growing body of knowledge on mycelium manipulation and open pathways for integrating MBCs into wider design practices where ecological responsibility and visual appeal are paramount. Follow-up studies should explore long-term feasibility, scalability, and integration of these techniques into architectural workflows.

## Figures and Tables

**Figure 1 biomimetics-10-00573-f001:**
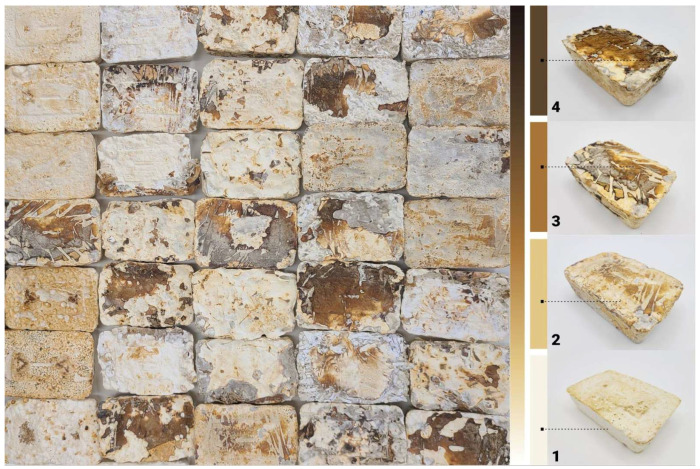
Mycelium skin colour variation observed during initial experiments. Top view of the bricks (**left**) and ¾ view of selected bricks showing different colours (**right**). The dimensions of each brick are 170 mm in length, 120 mm in width, and 55 mm in height.

**Figure 2 biomimetics-10-00573-f002:**
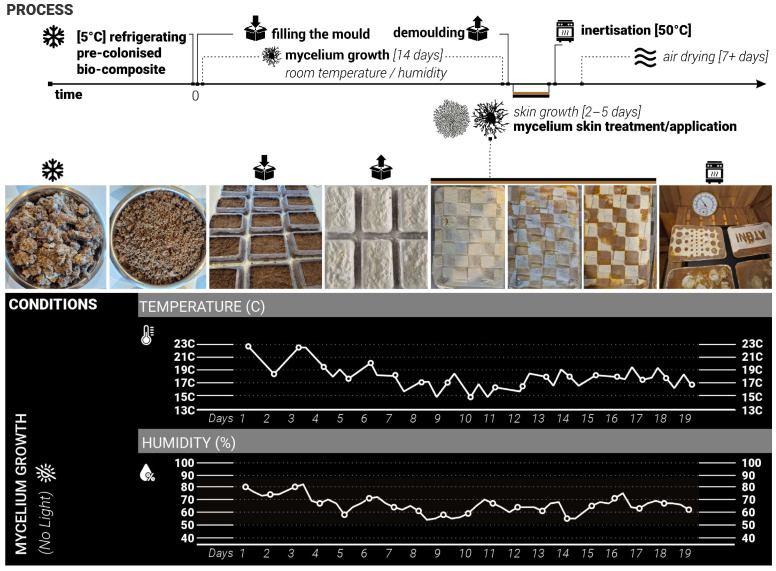
Growth process and conditions. **Top**: diagram showing the timeline for the process. **Middle**: pictures at each different stage, with the symbols corresponding to those in the timeline. **Bottom**: temperature and humidity as a function of days.

**Figure 3 biomimetics-10-00573-f003:**
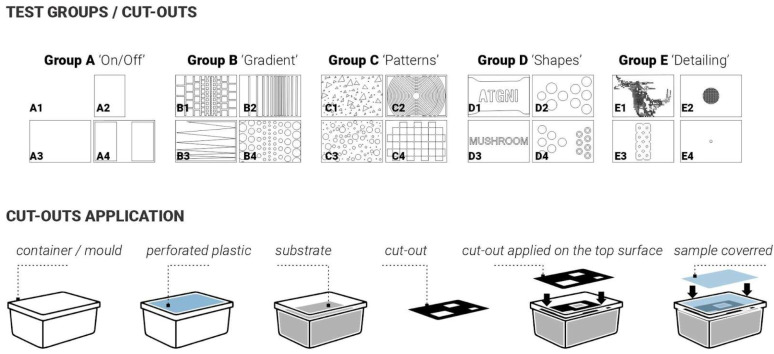
(**Top**): description of the five test groups (A, B, C, D, E) exploring different types of surface applications after demoulding. (**Bottom**): schematics representing the process for the application of the cut-outs.

**Figure 4 biomimetics-10-00573-f004:**
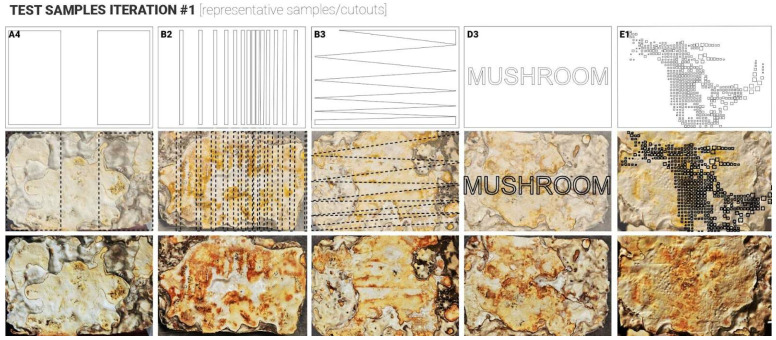
Top view of the samples produced during iteration 1 (representative samples/cutouts).

**Figure 5 biomimetics-10-00573-f005:**
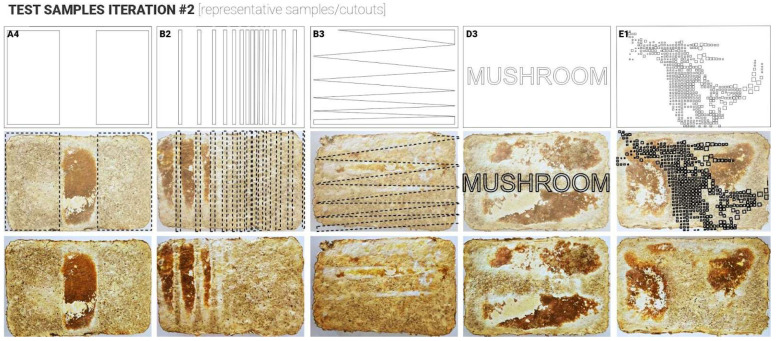
Top view of the samples produced during iteration 2 (representative samples/cutouts).

**Figure 6 biomimetics-10-00573-f006:**
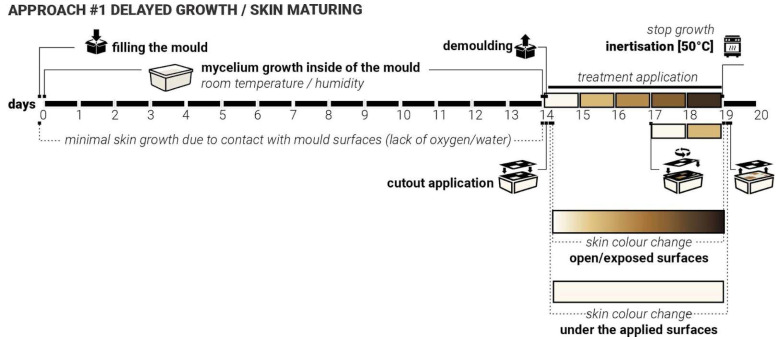
Approach #1 ‘Delayed Growth’, timeline, application and response.

**Figure 7 biomimetics-10-00573-f007:**
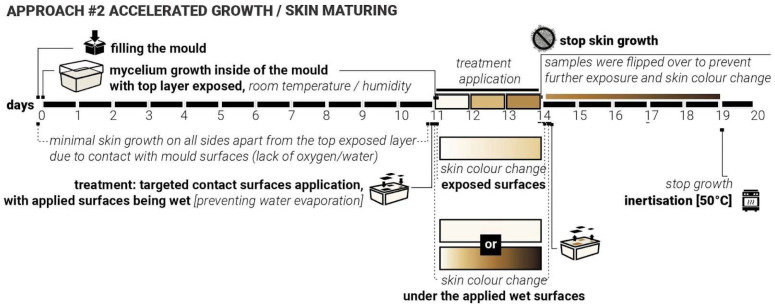
Approach #2 ‘Accelerated Growth’, timeline, application and skin response.

**Figure 8 biomimetics-10-00573-f008:**
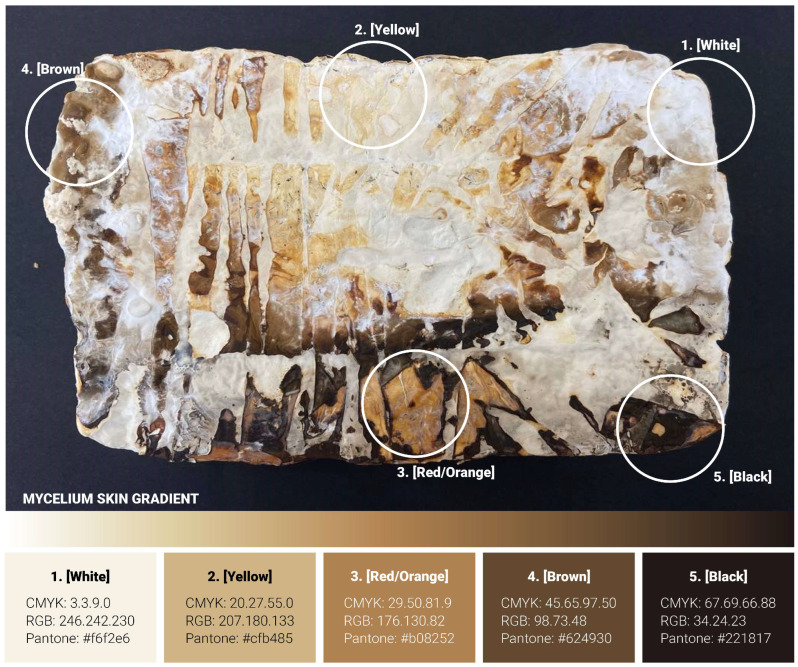
Top view of a sample with classification of colour variations in the mycelium skin (*Ganoderma Steyaertanum*). The five explored colour categories are: (1) White (Pantone #f6f2e6), (2) Yellow (Pantone #cfb485), (3) Orange (Pantone #b08252), (4) Brown (Pantone #f624930), (5) Black (Pantone #221817).

**Figure 9 biomimetics-10-00573-f009:**
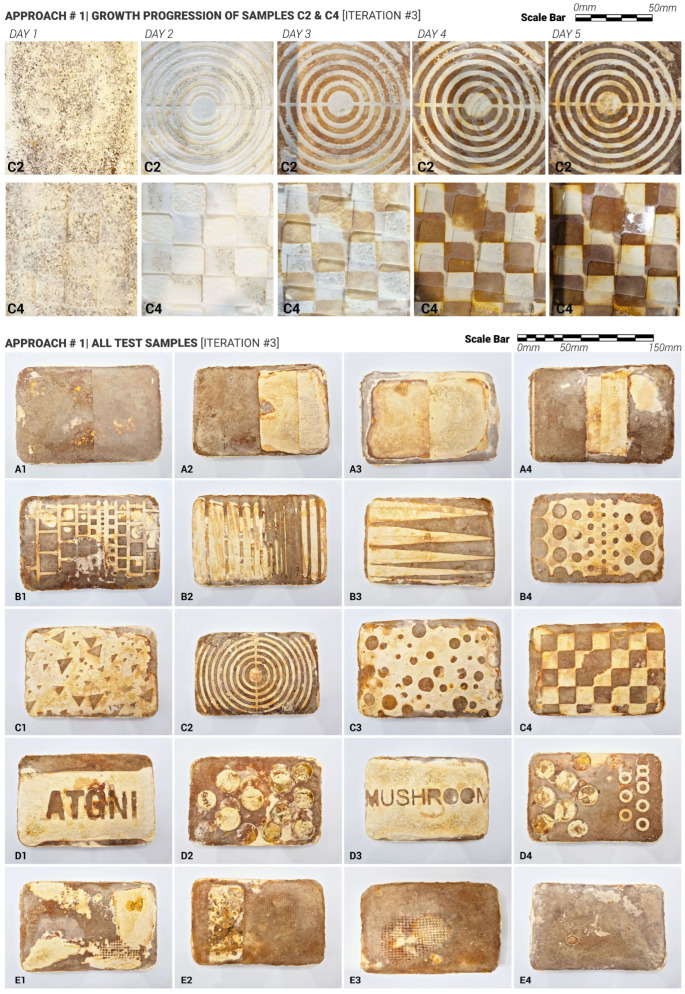
(**Top**): Top view of the samples produced following Approach 1, showing the growth progression from day 1 to day 5 of growth. (**Bottom**): Top view of all samples produced following Approach 1, with all the different cutouts tested. The dimensions of each sample (after drying) are 160 mm in length, 105 mm in width, and 35 mm in height.

**Figure 10 biomimetics-10-00573-f010:**
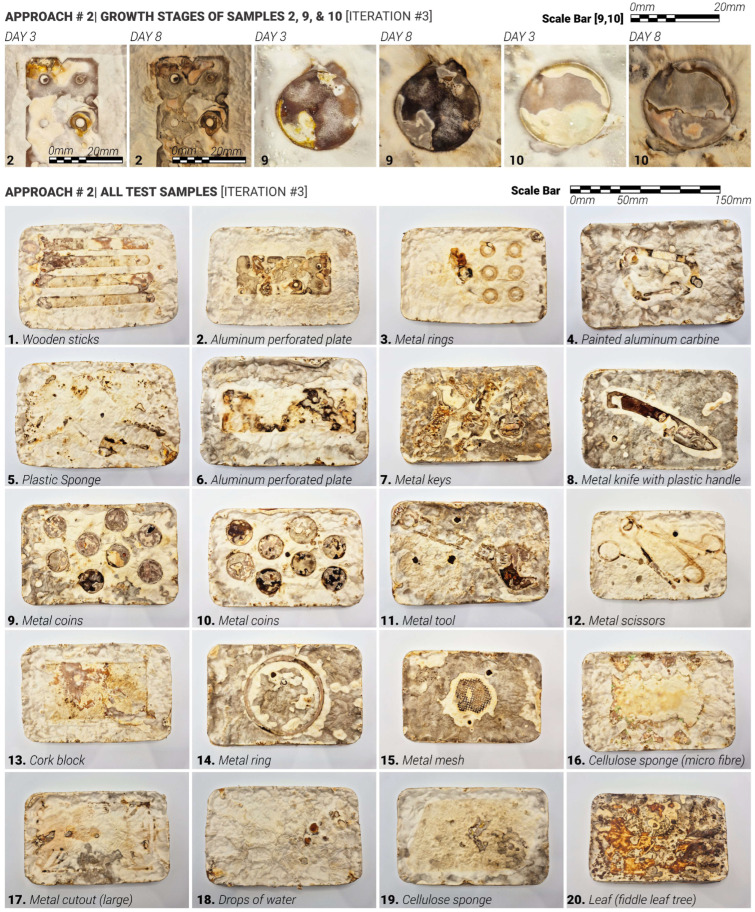
(**Top**): Top view of samples produced using Approach 2, showing the growth progression at day 3 and day 8 for 3 different designs. (**Bottom**): Top view of all the samples produced using Approach 2, using various materials. The dimensions of each sample (after drying) are 160 mm in length, 105 mm in width, and 35 mm in height.

**Figure 11 biomimetics-10-00573-f011:**
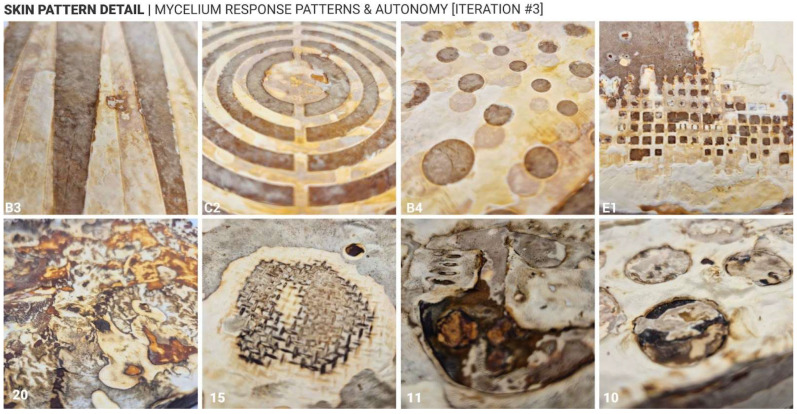
Mycelium skin detail & autonomy—response cases.

**Figure 12 biomimetics-10-00573-f012:**
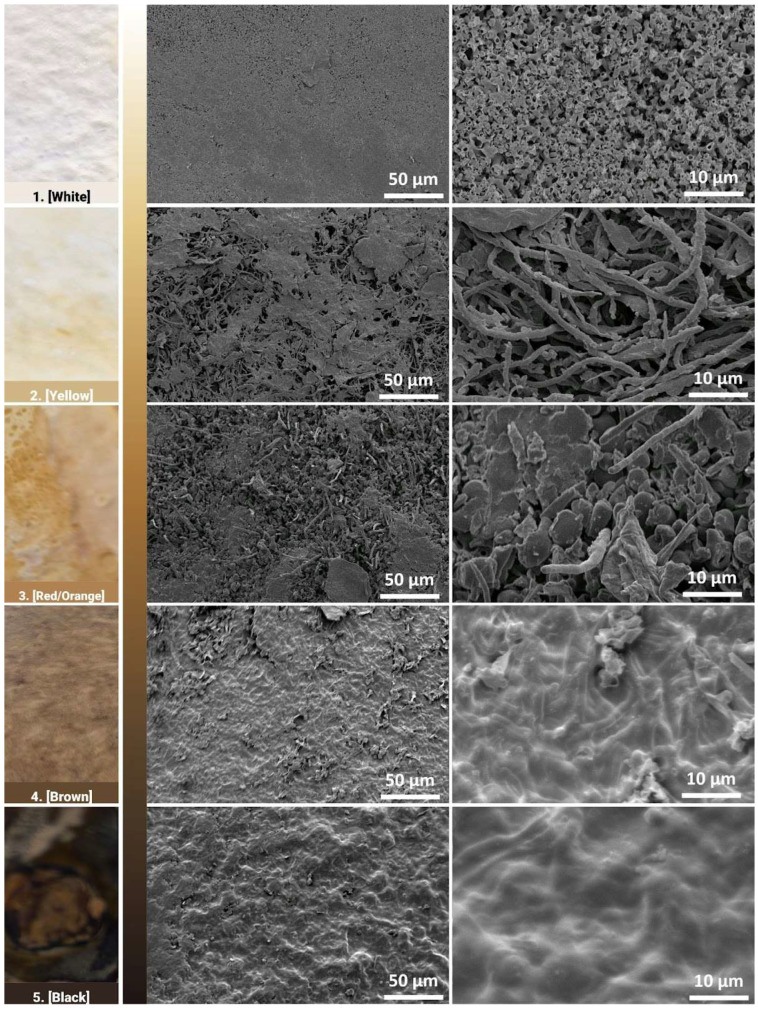
Electron micrographs of the fungal skin showing the surface morphology as a function of increasing brown colour.

**Figure 13 biomimetics-10-00573-f013:**
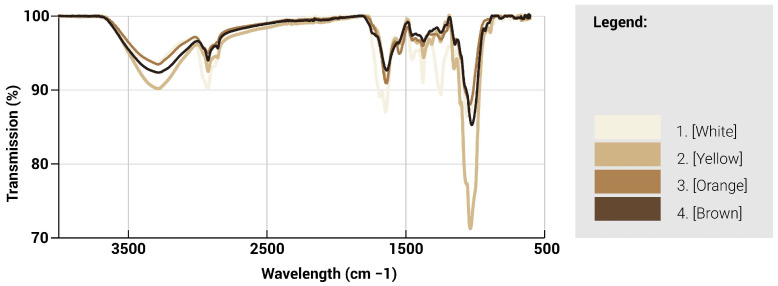
Transmission spectra as a function of wavelength for mycelium with different colours.

**Figure 14 biomimetics-10-00573-f014:**
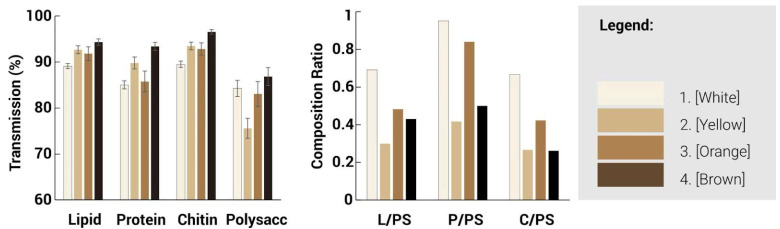
Transmission and composition ratio for mycelium skins of different colours showing the relative proportions of lipids (L), protein (P), chitin (C) and polysaccharides (PS).

**Figure 15 biomimetics-10-00573-f015:**
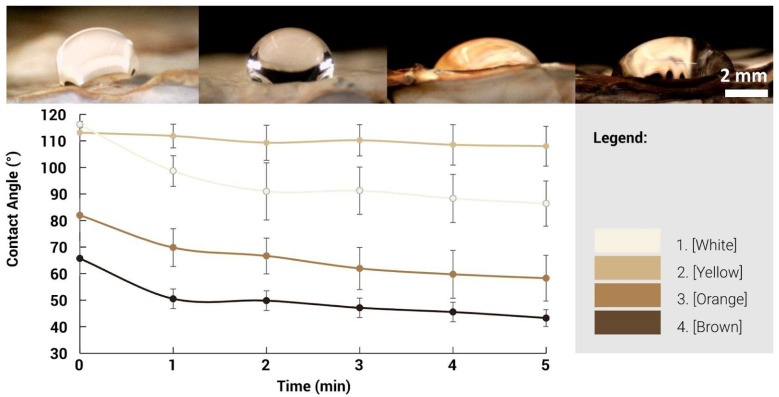
Top: photographs of water droplets deposited on mycelium skins with various colours. Bottom: Plot of the contact angle as a function of time for the various mycelium skins.

**Table 1 biomimetics-10-00573-t001:** Materials and Density.

Approach # 1	Application Material	Material Density
Cut-outs/Application	Plastic (Polypropylene)	910 (kg/m^3^)
**Approach # 2, Samples**	**Application Material**	**Material Density**
1 (Sticks)	Wood (Birchwood)	700 (kg/m^3^)
2, 6, 8 (Perforated Plate)	Aluminum	2710 (kg/m^3^)
3, 7, 8, 12 and 14 (Metal Tools), 15 (Mesh), 17 (Cutout)	Metal (Mild steel/Stainless steel)	7850–8000 (kg/m^3^)
9, 10 (Coins)	Metal (Copper)	8960 (kg/m^3^)
5 (Sponge)	Foamed plastic/Polyurethane	28 (kg/m^3^)
16, 19 (Sponge)	Cellulose sponge	24–28 (kg/m^3^)
13 (Block)	Cork (Standard density expanded)	115 (kg/m^3^)

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
