# Peer review of "Living Textures and Mycelium Skin Co-Creation: Designing Colour, Pattern, and Performance for Bio-Aesthetic Expression in Mycelium-Bound Composites"

_biomimetics, 2025, doi:10.3390/biomimetics10090573_

Round 1
Reviewer 1 Report
Comments and Suggestions for Authors
The question of aesthetically/functionally adapting mycelium-based panels/boards to the requirements of construction purposes appears of paramount importance as far as potential and diversified use is concerned.
The work is very novel and its structure is absolutely acceptable. I can only propose some improvements by inserting some more comments in the text:
- How was the selection of Ganoderma Steyaertanum (Reishi mushroom) for the purpose decided? On the basis of which considerations?
- How was the length of the growth/treatment cycle was decided? Was there some literature to support it?
- Since most of the work is concentrated on the change of color, how would be the effect in the long run with aging? Is there any way to control it?
- Also, the colors are classified in a very qualitative way. Is there a possibility in the future to offer some more quantitative classification (e.g., in a Pantone-like way)?
- The correlation between microstructure and color is also amazingly interesting. In the future, it would be interesting to understand whether and how much the biomass used for mycelium growth would act in the process of color passing from white to brown.
- ATR-FTIR study is also very interesting. Is there a way to correlate the color with the actual height of the various peaks, or the ratios of their intensities?
- i understand that literature on mycelium is not very abundant, but some attention to discussing the results in relation with further references would be appreciated.
Page 8 line 1: "another area of improvement. Shifting it from three to five days in the follow-up iterations". Please merge the two sentences, such as "another area of improvement, shifting it from three to five days in the follow-up iterations".
Reviewer 2 Report
Comments and Suggestions for Authors
The theme of this manuscript is the use of Reishi mycelium (Ganoderma Steyaertanum) to produce mycelium-bound composites (MBCs) with controllable surface colors and textures, and to achieve the designability of bio-aesthetics through two non-chemical treatment methods: Delayed Growth and Accelerated Growth. Its interdisciplinary integration of architectural design, materials science, and biology is evident, the research design is rigorous, and it is closely related to the application of sustainable building materials, making it very interesting.
This research is a pioneering exploration in the field of “aesthetic designability” of mycelium composites, establishing an example of cross-disciplinary research between materials science and design. However, I still have the following questions, so my recommendation before the manuscript can be published is Minor Revision.
First, please supplement the mechanical property and weather-resistance tests. The originally planned hardness and toughness tests (Vickers hardness, KIC) were not completed, resulting in insufficient evidence for the correlation between color/texture changes and structural performance. If feasible, please include tensile, bending, hardness, abrasion resistance, water absorption, UV aging, etc., to establish a comprehensive correlation between color–structure–performance.
The reproducibility of Accelerated Growth is poor and the variables in moisture control are complex. Color changes are greatly affected by the weight, density, and flatness of the contact materials, and in some samples the color is uneven or the effect is not obvious; the method of water application, contact time, and material absorbency have not been systematically controlled, increasing the variability of the experiment. It is recommended to use contact materials with standardized weight and flatness, precisely control the amount and time of water application to reduce uneven colors; please also evaluate the impact of natural coatings (plant oils, wax) on color saturation, durability, and water resistance, while maintaining degradability.
In addition, FTIR only shows limited differences in functional groups. It is speculated that the color is related to melanin deposition, but there is a lack of dedicated quantitative pigment analysis (such as HPLC, spectrophotometric quantification). It is recommended to use HPLC, mass spectrometry, or spectrophotometry to determine the content of melanin, carotenoids, etc., to analyze the chemical mechanism of color formation.
Finally, a few minor suggestions: for LINE 67–70, LINE 179–182, LINE 212–215, LINE 268–277, and LINE 324–325, please do not use bullet points; integrate them into narrative writing. For LINE 404–405, it is recommended to add this content to the caption of Figure 8 for easier reading; in Table 1 under Approach #2, please add explanations for each number for easier reading.
Reviewer 3 Report
Comments and Suggestions for Authors
The manuscript explores two post-demolding strategies-Delayed Growth (restricted Oâ‚‚/moisture via weighted cut-outs) and Accelerated Growth. The study documents iterative method refinements and couples visual outcomes with SEM, ATR-FTIR, and contact-angle measurements to relate color, microstructure, and surface properties. However, the current version of the manuscript would benefit from substantial revisions before it can be considered for publication.
Visual categories (white to black) need instrumental colorimetry and/or reflectance spectra with a color chart in every photograph for calibration. Provide per-sample color values over time for both approaches (day-by-day in Fig. 9; day 3 to 8 in Fig. 10).
Replace “weighted cut-outs” and “heavier objects” with contact pressure (kPa; mass/area) and report local RH/temperature at the interface (mini loggers or proxy measurements). Relate pigmentation to pressure, contact area fraction, porosity of contact material, and dwell time. Table 1 lists material densities—extend this to a regression/DOE linking pressure–porosity–moisture to color endpoints.
please add n, show individual data points for ATR-FTIR and contact angles, and apply appropriate tests to support claims that lighter regions are hydrophobic and darker regions hydrophilic. Include effect sizes and confidence intervals.
ATR-FTIR shows broad compositional trends but cannot resolve pigments. Add a targeted melanin assay / Raman band assignment / UV-Vis of extracted pigments to link darker tones to specific chromophores. Discuss limitations and outline validated proxy metrics (e.g., absorbance at characteristic wavelengths).
Provide explicit negative/positive controls: (1) demolded samples with no contact; (2) contact without added mass; (3) light-exposed vs dark to confirm authors’ assertion that color variation here is not light-driven.
Add scale bars to all macro images, include color charts, and specify camera/lighting/white-balance settings. Provide raw image sets, ImageJ scripts (contact angle, area analysis), and tabulated raw data in a public repository with DOIs.
